# Mechanical Properties of PLA Specimens Obtained by Additive Manufacturing Process Reinforced with Flax Fibers

Ana Paulo [1], Jorge Santos [1], João da Rocha [1], Rui Lima [2,3] and João Ribeiro [1,4,5,*]

1 Instituto Politécnico de Bragança, Campus de Santa Apolónia, 5300-253 Bragança, Portugal
2 METRICs, Mechanical Engineering Department, Campus de Azurém, University of Minho, 4800-058 Guimarães, Portugal
3 CEFT, Faculty of Engineering of the University of Porto (FEUP), Rua Dr. Roberto Frias, 4200-465 Porto, Portugal
4 Centro de Investigação de Montanha (CIMO), Instituto Politécnico de Bragança, Campus de Santa Apolónia, 5300-253 Bragança, Portugal
5 Laboratório Associado para a Sustentabilidade e Tecnologia em Regiões de Montanha (SusTEC), Instituto Politécnico de Bragança, Campus de Santa Apolónia, 5300-253 Bragança, Portugal
* Correspondence: jribeiro@ipb.pt

**Abstract:** Although polylactic acid (PLA) is one of the most used materials in additive manufacturing, its mechanical properties are quite limiting for its practical application, therefore, to improve these properties it is frequent to add fibers and, in this way, create a more resistant composite material. In this paper, the authors developed PLA composites reinforced with flax fibers to evaluate the improvement of tensile and flexural strength. The experimental design of experiments was based on the L18 Taguchi array where the control factors were the extruder temperature (three levels), number of strands (three levels), infill percentage of the specimens (three levels), and whether the flax fiber had surface chemical treatment. The tensile and flexural specimens were made on a 3D printing machine and was a mold was developed to fix and align the fiber strands during the printing process. The tensile and flexural experimental tests were performed in agreement with ASTM D638.14 and ISO 14125 standards, respectively. Analyzing the results, it was verified that the surface chemical treatment (NaOH) of the fiber did not show any influence in the mechanical properties of the composites; in contrast, the infill density demonstrated a huge influence for the improvement of mechanical strength. The maximum values of tensile and bending stress were 50 MPa and 73 MPa, respectively. The natural fiber reinforcement can improve the mechanical properties of the PLA composites.

**Keywords:** composite with natural fibers; flax fibers; PLA; additive manufacturing; mechanical properties

## 1. Introduction

In the last decade, additive manufacturing processes have been increasingly used in different applications [1] which can range from prototype manufacturing (first applications) [2] to industry [3,4], through leisure [5], to scientific research [6,7], among other applications [8,9]. There are many additive manufacturing processes [10]; however, one of the most common is the fused deposition modelling (FDM) [11]. FDM, also known as fused filament fabrication, is a process within the field of material extrusion. FDM employs filaments made of thermoplastic polymers and creates parts layer by layer by selectively de-positing melted material along a predetermined path [10]. The most popular thermo-plastic polymers used in FDM are the PLA (polylactic acid) and ABS (acrylonitrile butadiene styrene) [4]. ABS is a polymeric material that derives from petroleum and its composition has volatile organic compounds that can cause damage to the environment and human health [12]. On the other hand, PLA is a biodegradable polymer, highly interesting in technological terms due to its applications in the environmental field [13]. It is a type of

impact modified filament for the 3D printer, which is sustainable; it does not use volatile organic compounds, and allows the final product to have an accelerated degradation time through the action of humidity, temperature, light, and soil microorganisms, at the end of its useful life. As this is a thermoplastic polymer, which comes from renewable sugar-based raw materials, it may be an alternative to the use of non-biodegradable polymers or polymers with long-term degradation [14,15]. Despite the advantages of these materials, such as good adaptability to the FDM process, low cost and the obtained parts having a good resolution [16], the mechanical strength is relatively low [17] for more demanding applications. As a result, it would be interesting to improve some mechanical properties, namely, tensile and flexural strength, one possibility being to reinforce the parts with fibers (natural or synthetic).

Natural fibers are fibers that are not synthetic or manufactured. They come mainly from animals or plants [18]. Animal fibers consist of proteins (wool or silk), while plant fibers consist of cellulose [19]. Currently, natural fibers that originate from plants are the most widely used because they are suitable for use in composites with structural requirements. Moreover, plant fiber can be grown in many countries and can be harvested after short periods of time. In addition to cellulose, natural fibers are composed of lignin, hemicellulose, pectin, and waxes, and can be considered as natural composites containing mainly cellulose fibrils embedded in a lignin matrix. The nature of the cellulose and its crystallinity play an important role in reinforcing the efficiency of the natural fiber. The cellulose fibrils are aligned along the length direction of the fiber, ensuring maximum tensile and flexural strength, and providing stiffness [20]. Natural fibers have unique characteristics such as abundance, non-toxicity, high performance, versatility, and easy processing at low cost.

The natural fiber reinforced polymer composites (NFPCs) have several applications: besides the automotive industry, they arealso used in the construction industry due to their strength, low density, biodegradability, and high lifetime [21]. The fibers mostly used in industrial applications are flax, knaf and hemp because of the fibers' strength properties [21]. The properties of natural fibers vary, as they depend on the type of fiber, its source, and the moisture conditions. They depend on the fiber composition, the microfibril angle (i.e., angle of orientation of the micro-fibril in relation to the main fiber axis [22]), structure, defects, cell dimensions, physical properties, chemical properties, and also the fiber matrix [18]. Among the different plant natural fibers used in NFPCs, flax is one of the plant species that has been cultivated for longer in the world. It is a member of the genus Linum in the family Linaceae; Linum usitatissimum L. is the most common species among the 298 different species that are known. Flax fibers are found near the stem and are the mechanical support of the plant that is very thin [23–26]. They are considered one of the strongest fibers, because of their very complex structure [27]. These fibers are made up of a series of polyhedra that form overlapping elementary fibers over a considerable length, held together by an interface consisting mainly of hemicellulose and pectin. The typical diameter of an elemental flax fiber is between 10 and 15μm, although technical flax fibers range between 35 and 150μm [13].

As the need for FDM in industries grows over time, many researchers are drawn to it to improve the filler quality [28,29]. Synthetic or carbon fibers have often been used to reinforce the filler, although these fibers are harmful to the environment. Consequently, many researchers suggested using natural fiber instead of synthetic ones as the reinforcement, which can also be blended with a bio-polymer matrix, namely thermoplastics, as the polymer matrix in FDM industries. Multiple experimental tests have been performed to demonstrate the potential of natural fibers as the leading material in composite industries [30–34]. From these many pieces of research, it was possible to verify that there are some requirements which a natural fiber reinforced polymer composite (NFRPC) needs to fulfill to be manufactured by AM, namely (1) fiber homogeneity; (2) fiber alignment; (3) types of reinforcements and matrices; (4) adequate fiber-matrix bonding; (5) good interlayer bonding; and (6) minimal porosity [35]. Both the matrix materials, which hold

the fibers in place, and reinforcement need to be compatible with the selected 3D printing technique. Fiber distribution homogeneity is crucial to guarantee reliable properties throughout the printed part. The possibility to control fiber alignment and distribution in a predetermined location and direction enables the strengthening of sections of an object. Fiber reinforcement of proper length, size and shape must be selected to suit the intended purpose of the part. Fiber-loading is also essential for getting AM composites with good mechanical properties. A good interlayer fusion is necessary to prevent delamination. Ultimately the unwanted voids should be minimized because they would affect the mechanical properties of the NFRPC [35].

Based on these requirements, the authors of this paper developed a few experimental tests to determine an optimal combination of parameters to improve the mechanical properties of NFRPC made by FDM. To achieve the adequate fiber-matrix and interlayer bonding, many authors suggest the use of a surface chemical and physical treatment of fibers which enhances the adhesion properties of the interface between the fibers and the matrix, and decreases the absorption of water by the fibers [15,36,37]. These processes can be considered as modifiers of the properties of these fibers [38]. One of the most used chemical treatments is the alkaline one, in which the fibers are submerged in an alkaline solution, namely NaOH (sodium hydroxide), for a short period [39]. This increases the fiber's surface roughness and improves its mechanical properties. At the level of chemical bonding with the matrix material, it is possible to expose more cellulose on the fiber surface [39,40]. The most important change in this treatment is the breaking of the hydrogen bonds in the lattice structure, thus causing the surface roughness to increase. It also removes a certain amount of lignin, wax and oils that cover the outer surface of the fiber cell wall, exposing the short length crystals and depolymerizing the cellulose [41]. A consequence of a lower fiber-matrix bonding and interlayer bonding is the delamination. It was for this reason that the the influence of nozzle temperature was analyzed, or, in other words, the melting temperature of the PLA [35]. To evaluate the fiber homogeneity and alignment, long flax fibers were integrated into the specimen with the same direction of the applied load, varying the number of stands (fibers). According to Motru et al. [42], for PLA specimens reinforced with flax fibers, the ultimate tensile strength of the composite laminates tends to increase linearly with respect to the increase in weight percentage of the fiber. Therefore, specimens with three levels of fibers' percentage were tested and to guarantee the same position and distance of these fibers a mold was settled in which the fibers were fixed with glue tape. This type of fibers fixation has also another advantage that is the imposition of a pre-load in the fibers, despite the low value, that could improve the mechanical properties [43]. To evaluate the requirement of types of reinforcements and matrices, despite the materials referred to previously, it also depends on the raster orientation and infill level. The lowest ultimate tensile strength was reduced, proximally, by 55% in the specimen with the raster direction of 90° [44] and a higher infill density increases the tensile and yield strength [44].

The study presented in this paper aims to optimize the combination of flax fiber quantity, nozzle temperature, infill density and fiber surface chemical treatment to achieve the higher value of tensile and flexural strength of flax fiber reinforced composites.

## 2. Materials and Methods

### 2.1. Design of Experiments

Based on the literature referred to in the previous section, there are different manufacturing parameters that have a great influence on the properties of obtained products. Hence, the quality of the parts made by additive manufacturing are strongly influenced by the nozzle temperature and the infill density. In its turn, the mechanical properties of NFPCs vary considerably due to the fiber volume fraction (or number of strands) and the fiber surface treatment. For these reasons, the authors decided to use these parameters as control factors with different levels. In Table 1 the control factors with their respective levels are presented.

**Table 1.** Control factors of the composite.

| Symbol | Control Factor | Level 1 | Level 2 | Level 3 |
|--------|---------------|---------|---------|---------|
| A | Fiber surface treatment | NaOH treatment | No treatment | |
| B | Nozzle temperature | 190 °C | 200 °C | 220 °C |
| C | Number of strands | 10 | 15 | 20 |
| D | Infill density | 25% | 50% | 100% |

With the control factors and their respective levels previously defined, it was possible to create a Taguchi L18 orthogonal array (Table 2). This array was used for the implementation of both groups of experimental tests.

**Table 2.** Taguchi L18 orthogonal array.

| Test Number | A Fiber Surface Treatment | B Temperature | C Number of Strands | D Infill Density |
|-------------|---------------------------|---------------|---------------------|------------------|
| 1 | 1 | 1 | 1 | 1 |
| 2 | 1 | 1 | 2 | 2 |
| 3 | 1 | 1 | 3 | 3 |
| 4 | 1 | 2 | 1 | 1 |
| 5 | 1 | 2 | 2 | 2 |
| 6 | 1 | 2 | 3 | 3 |
| 7 | 1 | 3 | 1 | 2 |
| 8 | 1 | 3 | 2 | 3 |
| 9 | 1 | 3 | 3 | 1 |
| 10 | 2 | 1 | 1 | 3 |
| 11 | 2 | 1 | 2 | 1 |
| 12 | 2 | 1 | 3 | 2 |
| 13 | 2 | 2 | 1 | 2 |
| 14 | 2 | 2 | 2 | 3 |
| 15 | 2 | 2 | 3 | 1 |
| 16 | 2 | 3 | 1 | 3 |
| 17 | 2 | 3 | 2 | 1 |
| 18 | 2 | 3 | 3 | 2 |

## 2.2. Materials

### 2.2.1. Specimens Manufacturing

To determine the tensile and flexural strength, it was necessary to undergo tensile tests and flexural tests, according to the standards ASTM D638.14 [45] and ISO 14125 [46], respectively. Consequently, the geometries and dimensions of these specimens had to follow these standards. In Figure 1 the drawing of the two types of specimens is presented. Table 3 shows the values of specimen dimensions.

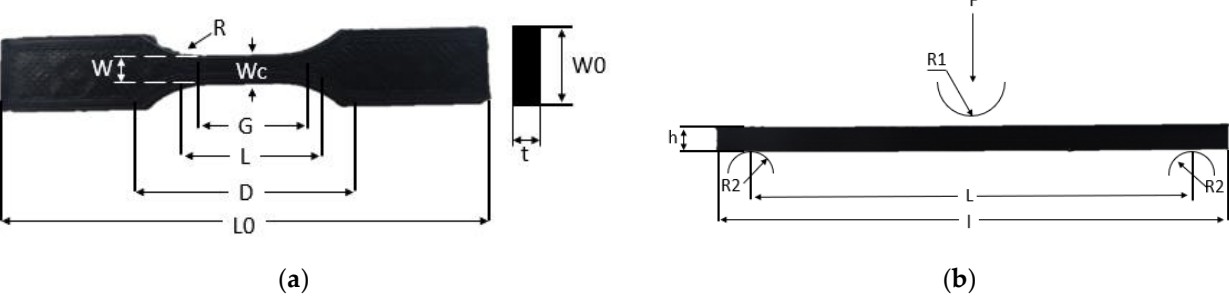

(**a**)  (**b**)

**Figure 1.** (**a**) Tensile specimen geometry (based on ASTM D638.14) and (**b**) Flexural specimen geometry (based on ISO 14125).

**Table 3.** Tensile and flexural specimen dimensions.

| Tensile Test Piece Dimensions [mm] | | Flexural Test Piece Dimensions [mm] | |
| :---: | :---: | :---: | :---: |
| D | 30 | F | - |
| G | 10 | h | 4 |
| L | 19 | I | 80 |
| L0 | 63 | L | 64 |
| R | 13 | R1 | 10 |
| t | 4 | R2 | 10 |
| W | 4 | | |
| W0 | 10 | | |
| WC | 4 | | |

The flax fibers are used to reinforce the 3D printed specimens and to manufacture the specimens the PLA (EasyFil PLA from FormFutura) which is a biodegradable thermoplastic was chosen. This material has other advantages, such as ease of printing, low deformation and good adhesion between layers. To spatially position the fibers in the defined locations, it was necessary to develop special molds. These molds were manufactured from wood (MDF) and boards were cut using a cutting laser machine (GCC X252). Figure 2 shows the molds used in this work (mold (a) used in tensile specimens and mold (b) used in flexural specimens). The fibers were attached to the grooves of the molds and the mold-fiber system was placed on the table of the 3D printer (Anycubic 3D printer) where the PLA deposition takes place.

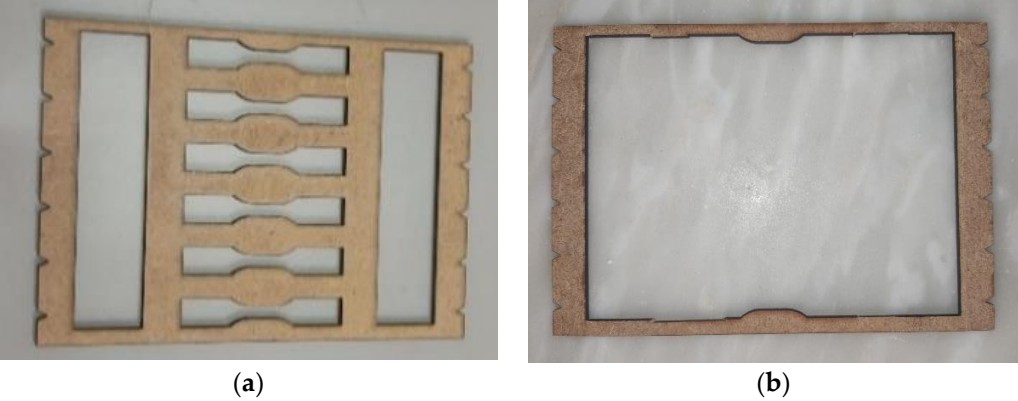

(**a**)　　　　　　　　　　　　　　　　　　　　　(**b**)

**Figure 2.** Molds used for tensile specimens (**a**) and flexural specimens (**b**).

The process began by joining each of the fibers with small knots, wrapping and attaching them to the grooves of the mold, according to the number of fiber strands that would be wanted, i.e., with 10, 15, or 20 strands. To improve the fixation of the fibers to the mold and guarantee their right position, pieces of tape (3 M) were used. This procedure also allowed us to apply a pre-load in the fibers.

The specimens manufacturing process begins with printing, approximately, half of specimen thickness directly on the printer table. When the printing reaches 50% of the thickness specimen, the printer is paused, and the mold with the fibers is placed in the middle of the printed specimen. Printing continues for the remaining 50% of the specimens. After the whole process, which takes about 2 h for the tensile tests and 4 h for the bending tests, it is necessary to let the samples and the machine cool down to remove the specimens (Figure 3a,b). After finishing the process, the 6 specimens from the tensile tests and the 12 specimens from the flexural tests are wrapped with adhesive tape.

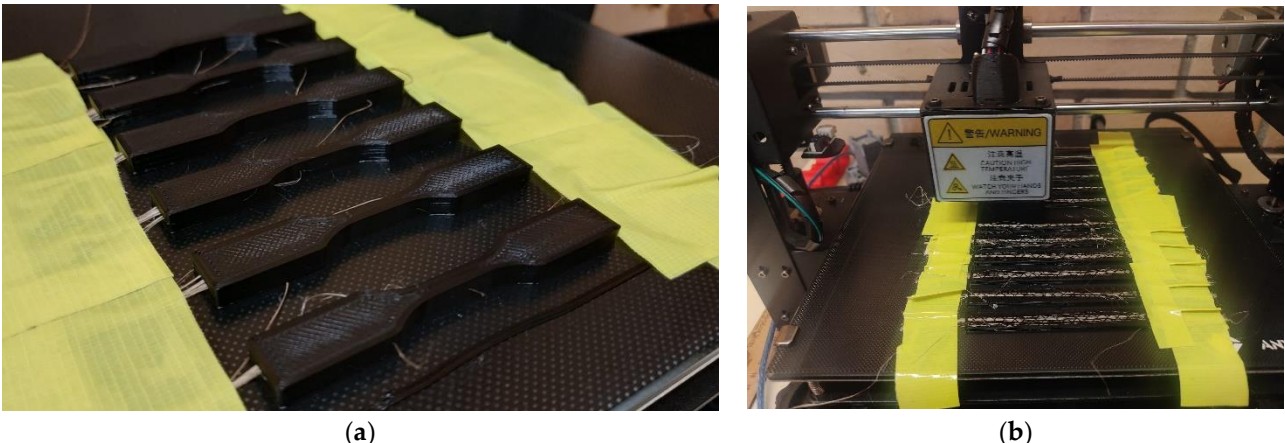

**Figure 3.** PLA printed specimens for tensile (**a**) and bending tests (**b**).

For the tensile specimens, 6 were made for each combination; thus, for the 18 combinations defined (L18), the total number of specimens for the tensile tests was 108. Likewise, for the bending tests, 6 specimens were produced for each combination; as such, 108 specimens were obtained for the 18 combinations. To evaluate the improvement or decrease of mechanical properties, PLA specimens without any reinforcement were manufactured, and, consequently, three groups of specimens with 25%, 50% and 100% of infill density were printed. For each group, 6 specimens were made which makes a total of 18.

### 2.2.2. Chemical Surface Treatment

To improve the surface properties of the flax fibers, they were subjected to a sodium hydroxide treatment. The flax fibers were immersed in a 5% sodium hydroxide (NaOH) solution for 3 h at room temperature. After this treatment, the fibers were washed with 5% acetic acid to neutralize the NaOH. To remove all residues of acetic acid, the flax fibers were washed with distilled water and then they were dried in a 120 °C oven (Scientific Series 9000 Oven) for 2 h. The result from drying at room temperature (approximately 20 °C) for 24 h can be seen in Figure 4b.

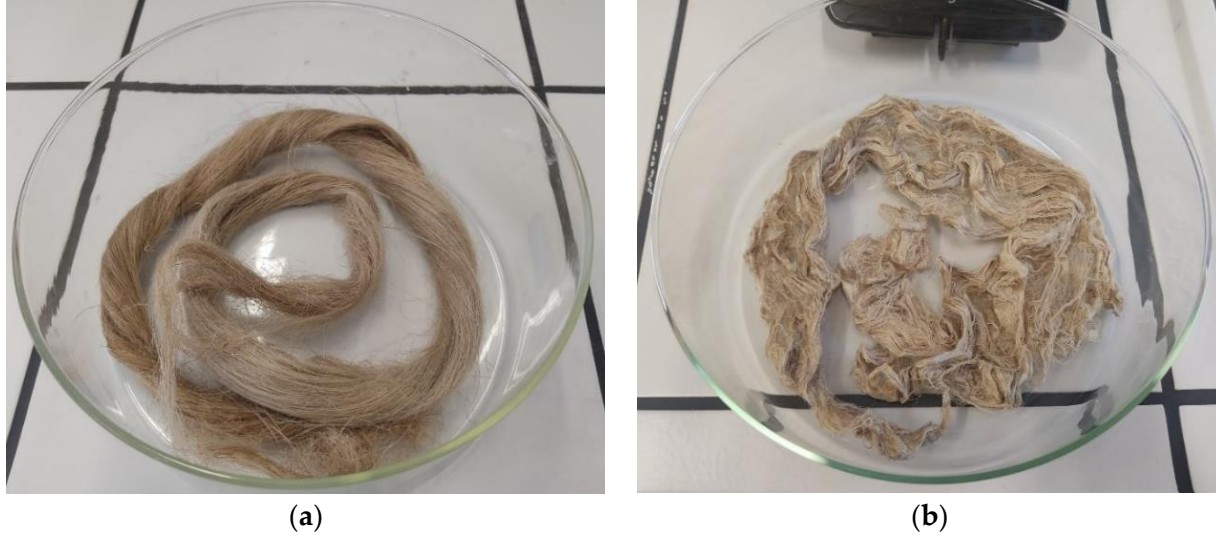

**Figure 4.** Untreated flax (**a**) and treated flax (**b**).

### 2.3. Tensile Tests

The tensile test took place in the laboratory of structures and strength of materials. To perform the experimental test the ASTM D638.14 [45] standard was used, in which the test

speed was 10 mm/min. Thus, a specimen (Figure 5a) was inserted in the machine, placed at 30 mm between the clamps (Figure 5b).

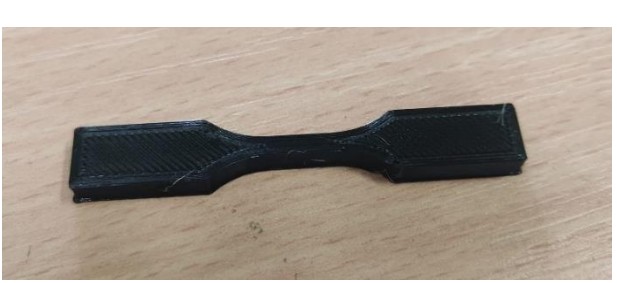

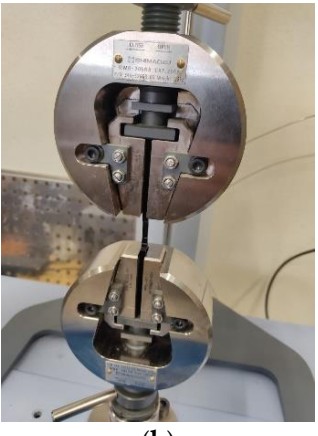

(**a**) (**b**)

**Figure 5.** Tensile specimen (**a**) and tensile tests (**b**).

The tensile strength can be calculated from the maximum load and the transverse area. To represent the stress-strain curve, it is necessary to determine the respective values, for each point. The stress is calculated by dividing the applied load by the average cross-sectional area.

### 2.4. Flexural Tests

For this experimental work the three-point flexural test was used (Figure 6).

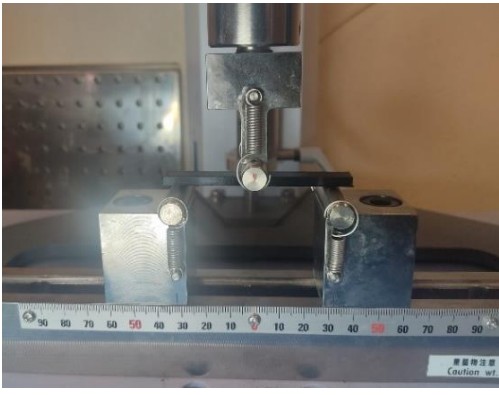

**Figure 6.** Bending test.

The standard used was ISO 14125 [46], as well as a test speed at 1.7 mm/min. The test speed was calculated using the Equation (1) [46]:

$$V = \frac{\varepsilon' L^2}{6h} \tag{1}$$

where:

V: Test speed (mm/min)
$\varepsilon'$: Deformation rate of 1%/min
L: Outer span (mm)
h: Thickness (mm)

Thus, the specimen was inserted in the machine according to the standard ISO 14125. After finishing all the tests, it was possible to check the graphs and the obtained data.

For the three-point flexural test (Figure 6), it was necessary to use a test specimen, that is, a specimen in the shape of a flat beam of constant rectangular cross section. In three-point

flexural, the maximum bending stress occurs at the outer surface of the specimen and is given by Equation (2) [46]:

$$\sigma = \frac{3PL}{2bh^2} \tag{2}$$

where:

σ: Bending stress at the outer surface (MPa)
*P*: Load applied at the point (N)
L: Specimen length (mm)
b: Specimen width (mm)
h: Specimen thickness (mm))

The strain is calculated from the Equation (3) [46]:

$$\varepsilon = \frac{6\delta h}{L^2} \tag{3}$$

where:

ε: Deformation
δ: Deflection—Distance of the lower or upper surface of the specimen in the middle of the span that has deviated from the initial position (mm)
h: Specimen thickness (mm)
L: Specimen length (mm)

### 3. Results and Discussion

After analyzing the results obtained through the universal testing machine, the stress and strain were computed, and a graph was drawn for each of the 108 tests for tensile and flexural tests. In addition to these tests, the same was done for the specimens without fibers, with a temperature of 200 °C and a fill percentage of 50%, corresponding to test 19* (for the tensile and flexural tests). An analysis of each graph was done, and the following graphs of the averages for each test were obtained.

Figure 7 presents the stress-strain curve for tensile tests. Each curve (ex. Test 1, Test 2, etc.) represents the average of the six specimens for each L18 combination.

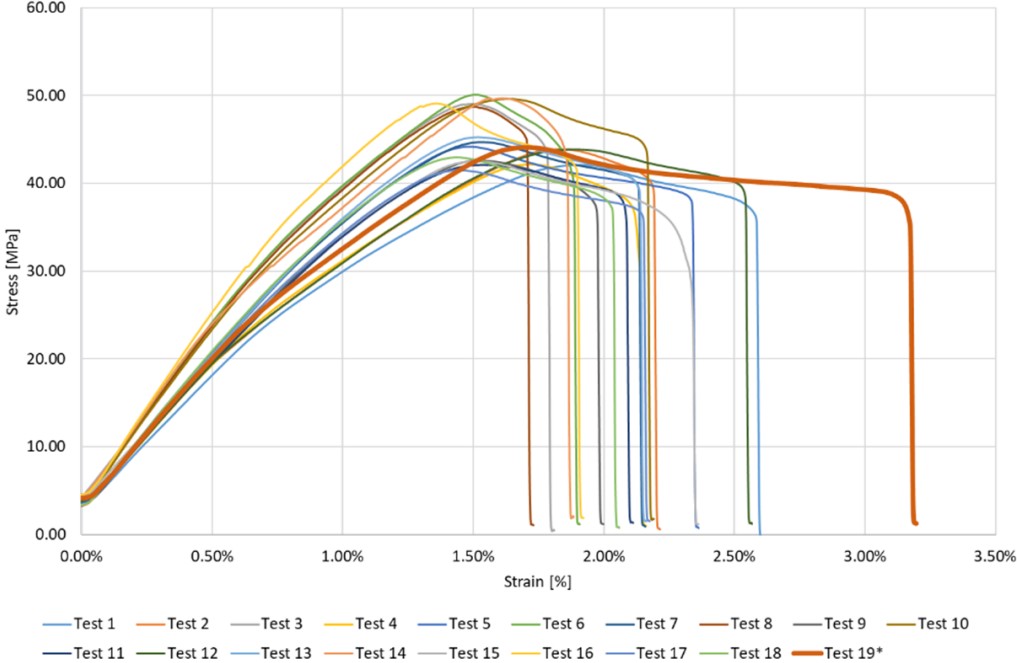

**Figure 7.** Stress-strain for tensile strength.

Figure 7 shows that test 6 achieved the best results with a maximum tensile stress, followed by tests 3 and 8. First, test 6 represents the chemically treated specimens, at a temperature of 200 °C, with 20 fibers and an infill percentage of 100%. Secondly, test 10 depicts the specimens without chemical treatment, at a temperature of 190 °C, with 10 fibers and an infill percentage of 100%. Finally, test 14 represents the specimens without chemical treatment, at a temperature of 200 °C, with 15 fibers and an infill percentage of 100%. In other words, both tests share the fact that the infill percentage is total (100%).

Figure 8 presents the stress-strain curve for flexural tests. Each curve represents the average of the six specimens for each combination defined by the L18 Taguchi array.

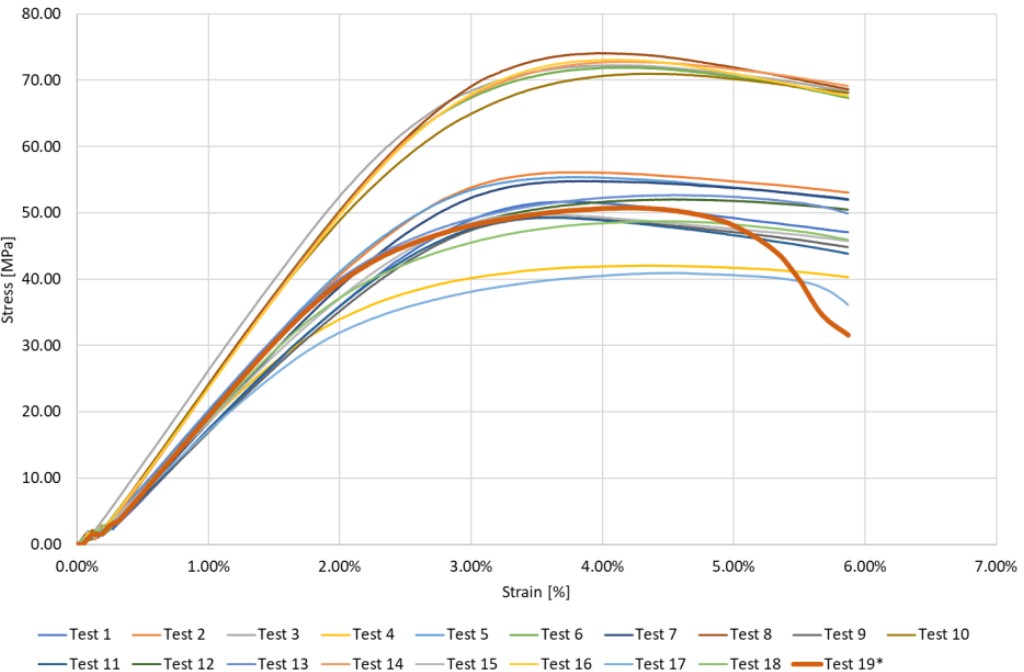

**Figure 8.** Stress-strain for flexural strength.

Figure 8 shows that test 8 revealed the best results, followed by tests 16 and 14. To begin with, test 8 represents the chemically treated specimens, at a temperature of 220 °C, with 15 fibers and an infill percentage of 100%. Then, test 16 represents the specimens without chemical treatment, at a temperature of 220 °C, with 10 fibers and an infill percentage of 100%. Finally, test 14 represents the specimens without chemical treatment, at a temperature of 200 °C, with 15 fibers and an infill percentage of 100%. This means that both tests only share the percentage of infill, which was the maximum.

Table 4 shows the mean values of the maximum stresses for the 6 specimens in each test, as well as their standard deviation, the last line of this table (experiment 19*) represents the experimental test for specimens with pure PLA. As represented graphically, it can be seen that essay 6 obtained the highest ultimate strength for the tensile test which has the value of 49.96 MPa and a standard deviation of 0.89 MPa. For the bending tests, the one with the highest stress was test 8 with a value of 72.94 MPa and a standard deviation of 2.09 MPa. In both cases, the values of standard deviation are very low which is an indication that all specimen groups have a very similar behavior to the average. Related to the test performed without fibers, the tensile test obtained a value of 43.43 MPa with a standard deviation of 1.33 MPa. For the bending, it achieved a value of 53.55 MPa and a standard deviation of 4.59 MPa. These results demonstrate that, with the reinforcement of natural fibers, the composite becomes more resistant.

**Table 4.** Values of the mean maximum stresses and their standard deviation.

| Experiment Number | Average Tensile Strength [MPa] | Standard Deviation [MPa] | Average Flexural Strength [MPa] | Standard Deviation [MPa] |
|---|---|---|---|---|
| 1 | 41.99 | 0.72 | 51.04 | 0.78 |
| 2 | 44.02 | 0.52 | 54.13 | 3.38 |
| 3 | 48.25 | 1.59 | 72.59 | 0.79 |
| 4 | 41.99 | 0.49 | 44.32 | 3.96 |
| 5 | 44.01 | 0.70 | 54.22 | 2.79 |
| 6 | 49.96 | 0.89 | 71.86 | 1.13 |
| 7 | 44.55 | 1.02 | 52.80 | 3.52 |
| 8 | 49.16 | 0.88 | 72.94 | 2.09 |
| 9 | 41.86 | 1.09 | 46.71 | 3.68 |
| 10 | 49.55 | 0.68 | 70.97 | 0.50 |
| 11 | 42.11 | 0.50 | 46.30 | 3.81 |
| 12 | 44.41 | 0.59 | 53.57 | 3.05 |
| 13 | 45.03 | 0.83 | 54.08 | 2.85 |
| 14 | 49.25 | 1.02 | 72.13 | 1.71 |
| 15 | 42.69 | 0.54 | 48.89 | 2.75 |
| 16 | 48.85 | 0.65 | 72.41 | 1.69 |
| 17 | 41.09 | 0.48 | 41.78 | 2.75 |
| 18 | 43.26 | 1.37 | 50.97 | 3.82 |
| 19 * | 43.43 | 1.33 | 53.55 | 4.59 |

Analyzing the results shown in Table 4, is possible to observe that the fibers can improve or worsen mechanical strength (tensile and flexural). Thus, the average tensile strength for the specimens without fibers is 43.43 MPa while the best result for the NFPCs is 49.96 MPa (Test 6), however, for some tests, like test 17, the tensile strength decreases (41.09 MPa). The same observation happened for the flexural tests, the average flexural strength of the pure PLA specimen is 53.55 MPa and for test 3 the value is 72.59 MPa; still, for test 17 the flexural strength is 41.78 MPa.

The experimental results can be converted into a signal-to-noise ratio (S/N). Taguchi suggests using the S/N ratio to determine the quality characteristics that deviate from the desired values.

The Signal-to-Noise (S/N) ratio developed by Taguchi is a performance measure for choosing the control levels that best handle noise. In this method the term "signal" symbolizes the desired value for the output characteristic, and the term "noise" symbolizes the undesired value. The Signal-to-Noise ratio takes into account the mean and variance, that is, it is the ratio between the mean (signal) and the standard deviation (noise). The S/N equation depends on the criteria for the quality characteristic to be optimized. There are three categories of the quality characteristics in the S/N ratio analysis, which are the lowest better, the highest better and the nominal best. Regardless of the quality category of the features, a higher S/N ratio will correspond to better quality features. Thus, the level with the highest S/N ratio is the optimal level of the control factors. With the analysis of variance, it is possible to see which control factors are statistically feasible, using the results obtained in the S/N and ANOVA analyses, the optimal combination of control factors and the prediction of their levels. The goal of this study is to maximize tensile and flexural strength, thus, the quality feature category for the S/N ratio is the highest-best:

$$S/N = -10\log\left(\frac{1}{n}\sum_{i=1}^{n}\frac{1}{y_i^2}\right) \tag{4}$$

where n is the number of observations and $y_i$ are the observed data [20,47].

The S/N ratios for tensile and flexural strength are shown in Table 5.

**Table 5.** S/N ratios for tensile and flexural strength.

| Test Number | A | B | C | D | S/$N_{ts}$ Ratio [db] | S/$N_{fs}$ Ratio [db] |
|---|---|---|---|---|---|---|
| 1 | 1 | 1 | 1 | 1 | 32.46 | 34.15 |
| 2 | 1 | 1 | 2 | 2 | 32.87 | 34.61 |
| 3 | 1 | 1 | 3 | 3 | 33.66 | 37.22 |
| 4 | 1 | 2 | 1 | 1 | 32.46 | 32.84 |
| 5 | 1 | 2 | 2 | 2 | 32.87 | 34.65 |
| 6 | 1 | 2 | 3 | 3 | 33.97 | 37.13 |
| 7 | 1 | 3 | 1 | 2 | 32.97 | 34.39 |
| 8 | 1 | 3 | 2 | 3 | 33.83 | 37.25 |
| 9 | 1 | 3 | 3 | 1 | 32.43 | 33.30 |
| 10 | 2 | 1 | 1 | 3 | 33.90 | 37.02 |
| 11 | 2 | 1 | 2 | 1 | 32.49 | 33.22 |
| 12 | 2 | 1 | 3 | 2 | 32.95 | 34.54 |
| 13 | 2 | 2 | 1 | 2 | 33.07 | 34.63 |
| 14 | 2 | 2 | 2 | 3 | 33.84 | 37.15 |
| 15 | 2 | 2 | 3 | 1 | 32.61 | 33.74 |
| 16 | 2 | 3 | 1 | 3 | 33.78 | 37.19 |
| 17 | 2 | 3 | 2 | 1 | 32.27 | 32.37 |
| 18 | 2 | 3 | 3 | 2 | 32.71 | 34.07 |

In Table 5, S/Nts is the S/N ratio for tensile strength and S/Nfs is the S/N ratio for flexural strength, where the S/N results for the 18 combinations are represented.

For a higher S/N ratio, the best category is applied in order to maximize the response (tensile and flexural strength). The average S/N ratio for the control factors of levels 1, 2 and 3 can be calculated by averaging the S/N ratios of the corresponding tests. In Tables 6 and 7, the average S/N ratio for each level of control factor, i.e., the response, is shown.

**Table 6.** Mean response table of S/N ratio for tensile strength and significant interaction.

| Symbol | Control Factor | Mean S/N Ratio [db] | | |
|---|---|---|---|---|
| | | Level 1 | Level 2 | Level 3 |
| A | Fiber surface treatment | 33.06 | 33.07 | |
| B | Nozzle temperature | 33.05 | 33.14 | 33.00 |
| C | Number of strands | 33.11 | 33.03 | 33.05 |
| D | Infill density | 32.45 | 32.91 | 33.83 |

**Table 7.** Mean response table of the S/N ratio for flexural strength and significant interaction.

| Symbol | Control Factor | Mean S/N Ratio [db] | | |
|---|---|---|---|---|
| | | Level 1 | Level 2 | Level 3 |
| A | Fiber surface treatment | 35.06 | 34.88 | |
| B | Nozzle temperature | 35.13 | 35.02 | 34.76 |
| C | Number of strands | 35.04 | 34.88 | 35.00 |
| D | Infill density | 33.27 | 34.48 | 37.16 |

Figures 9 and 10 show the response graph of the S/N ratio for tensile and flexural strength, respectively.

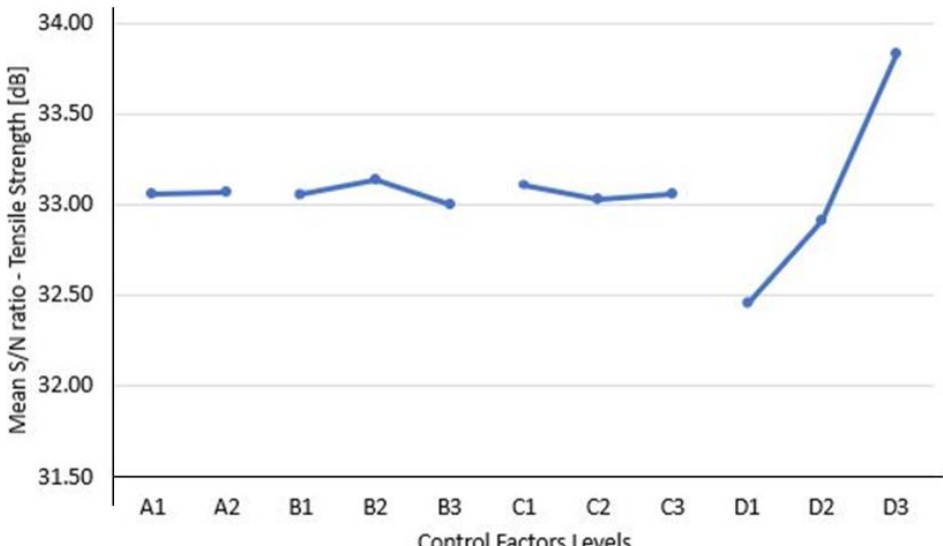

**Figure 9.** S/N ratio response graph for tensile strength.

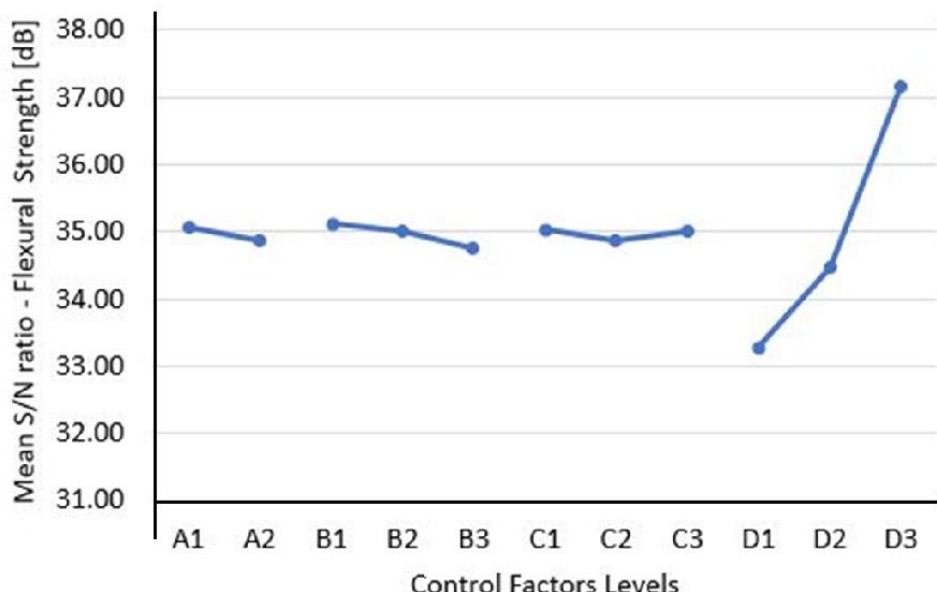

**Figure 10.** S/N ratio response graph for flexural strength.

Through the analysis of Figure 9, it is possible to determine the optimal combination to increase the tensile strength of the flax fiber reinforced composite. Regarding the tensile tests, the ideal combination is A1/A2, B2, C1 and D3, i.e., with/without chemical treatment, with a temperature of 200 °C, 10 fibers and an infill percentage of 100%.

By analyzing Figure 10, it is possible to determine the optimal combination to increase the flexural strength of the flax fiber reinforced composite. Concerning the bending tests, the optimal combination is A1, B1, C1 and D3, i.e., with chemical treatment, a temperature of 190 °C, 10 fibers and an infill percentage of 100%.

The relative importance of the control factors for tensile and flexural strength needs to be identified more precisely using ANOVA analysis to determine the levels of importance of the control factors. In Tables 8 and 9, it is clear which control factor influences more the tensile and flexural strength values.

**Table 8.** ANOVA for tensile strength.

| Source | DF | Adj SS | Adj MS | F-Value | *p*-Value | Contribution [%] |
|---|---|---|---|---|---|---|
| A | 1 | 0.00 | 0.00 | 0.00 | 0.96 | 0.00% |
| B | 2 | 0.07 | 0.03 | 2.85 | 0.11 | 1.12% |
| C | 2 | 0.01 | 0.01 | 0.57 | 0.58 | 0.22% |
| D | 2 | 5.80 | 2.90 | 245.99 | 0.00 | 96.69% |
| Error | 10 | 0.12 | 0.01 | | | 1.97% |
| Total | 17 | 6.00 | | | | 100.00% |

**Table 9.** ANOVA for flexural strength.

| Source | DF | Adj SS | Adj MS | F-Value | *p*-Value | Contribution [%] |
|---|---|---|---|---|---|---|
| A | 1 | 0.14 | 0.14 | 0.88 | 0.37 | 0.29% |
| B | 2 | 0.43 | 0.21 | 1.31 | 0.31 | 0.85% |
| C | 2 | 0.08 | 0.04 | 0.26 | 0.78 | 0.17% |
| D | 2 | 47.52 | 23.76 | 146.00 | 0.00 | 95.42% |
| Error | 10 | 1.63 | 0.16 | | | 3.27% |
| Total | 17 | 49.80 | | | | 100.00% |

In Tables 8 and 9, the variance results for each control factor can be checked, where DF is the degree of Freedom, Adj SS is the sum of squares and Adj MS are the mean squares. The F-test is a statistical tool to check which parameters significantly affect the quality of the characteristics, that is, it is defined as the ratio of the mean square deviations to the mean square error.

After analyzing the results of the F-test value, it is possible to verify that, for tensile and flexural strength, the most significant control factors are the nozzle temperature and the infill percentage of the specimen, with the remaining factors varying. For tensile strength, the infill percentage was very significant with 96.69%, followed by the nozzle temperature, number of strands, and the use or not of fiber treatment, with 1.12%, 0.22%, and 0%, respectively. For flexural strength, the infill percentage was also the most significant with 95.42%, followed by the nozzle temperature, the use or not of fiber treatment, and the number of strands with 0.85%, 0.29% and 0.17%, respectively.

Analyzing these results, is possible to observe that the most important parameter that contributes to the improvement of studied mechanical properties is the infill density (or percentage), because the increase of this parameter is directly related to the growth of the specimen strength and resistance. On other hand, the influence of the fibers and their chemical surface treatment is very low. The reason for this happening might be related to the adhesion between the PLA and the fibers. The temperature to ensure that the PLA has sufficient fluidity to cover the entire outer area of the fiber is not high enough, whereas molten PLA does not allow good wettability to fill the entire fiber.

As the optimal combinations of parameters and levels are different from the L18 analysis, confirmation tests are required.

*Confirmation Tests*

Given the results obtained, the tests were performed again for the optimal combinations, i.e., for the tensile tests, 3 specimens were made without chemical treatment, with a nozzle temperature of 200 °C, 10 fibers, and an infill percentage of 100%. For the bending tests, 3 specimens were also made, but with chemical treatment, a nozzle temperature of 190 °C, 10 fibers, and an infill percentage of 100%. Only 3 specimens were produced, because the standard deviation of the previous tests was quite small. Figure 11 graphically represents the optimal combinations of tensile strength.

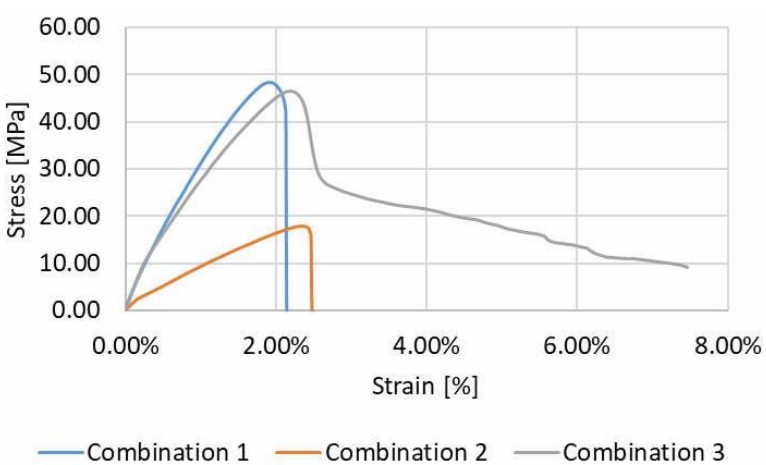

**Figure 11.** Optimum combinations of tensile strength.

A maximum stress of 48.43 MPa can be observed, i.e., it is very close to the stresses of the best tests performed previously. Figure 12 graphically represents the optimal combinations of flexural strength.

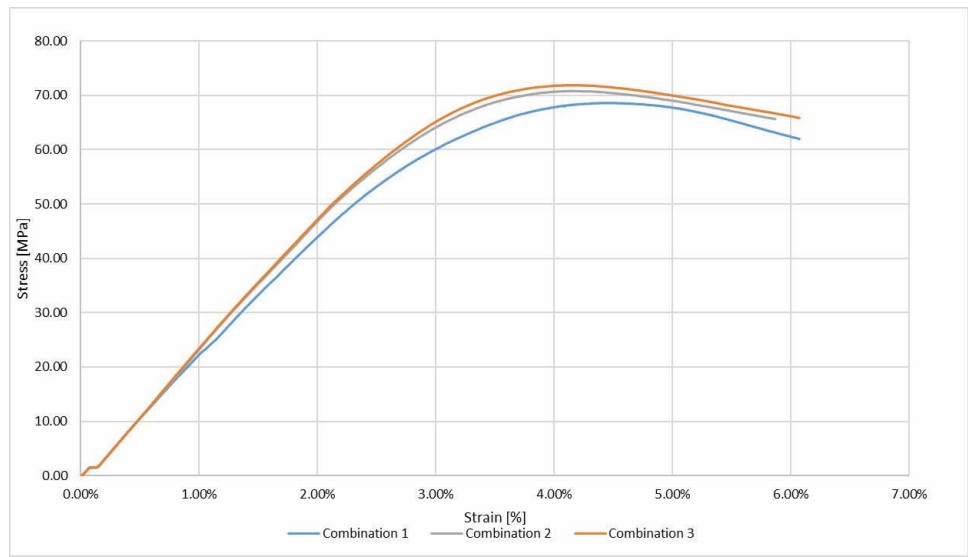

**Figure 12.** Optimum combinations of flexural strength.

A maximum stress of 71.90 MPa can be observed, which is quite similar to the best tests performed previously.

## 4. Conclusions

The main conclusions of this work are as follows:

- The maximum stress for the tensile test is 49.96 MPa and for the bending is 72.94 MPa with a standard deviation of 0.89 MPa and 2.09 MPa, respectively. The values of the standard deviation are very low which is a sign that all specimens within groups have a very similar behavior. This observation could be an interesting feature for industrial applications, because it is possible to guarantee very similar properties for manufactured products by this process.
- The natural fiber reinforcement, for many combinations of control factors, improve the mechanical strength of the composite. Comparing the tensile strength of pure PLA with best result of the composite the values are 43.43 MPa and 49.96 MPa, respectively. The flexural test obtained a value of 53.55 MPa (PLA without fibers) compared to the maximum value of the flax fiber reinforced composite of 72.94 MPa. However, the

pure PLA can be stronger than some combinations of NFPCs, thus, for example, the test 17 showed low values of mechanical strength (tensile and flexural) (41.09 MPa and 41.78 MPa, respectively).

- With the Taguchi method, it was possible to determine the optimal combinations for maximum tensile and flexural strength. For tensile, the optimum combination is A1/A2, B2, C1 and D3, i.e., with/without chemical treatment, with a nozzle temperature of 200 °C, 10 fibers and an infill percentage of 100%. For bending, the optimal combination is A1, B1, C1 and D3, i.e., with chemical treatment, a nozzle temperature of 190 °C, 10 fibers and infill percentage of 100%.
- With the analysis of variance, it was found that the infill percentage was the parameter with the greatest contribution to the increase of tensile and flexural strength with percentages of 96.69% and 95.42%, respectively.
- After performing the confirmation tests with the optimal combinations obtained previously, the maximum values for the tensile and flexural strength were determined at 48.43 MPa and 71.90 MPa, respectively.

**Author Contributions:** Conceptualization, J.R. and J.d.R.; Data curation, A.P. and J.S.; Investigation, A.P., J.S. and R.L.; Methodology, A.P., J.S., J.d.R. and J.R.; Resources, J.R., J.d.R. and R.L.; Software, A.P.; Supervision, J.d.R., R.L. and J.R.; Validation, A.P. and J.S.; Writing—original draft, A.P., J.S., J.R. and J.d.R.; Writing—review and editing, A.P. and J.R. All authors have read and agreed to the published version of the manuscript.

**Funding:** Financial support was provided by Portugal's national funding FCT/MCTES (PIDDAC) to Centro de Investigação de Montanha (CIMO) (UIDB/00690/2020 and UIDP/00690/2020) and SusTEC (LA/P/0007/2020).

**Data Availability Statement:** Not applicable.

**Conflicts of Interest:** The authors declare no conflict of interest.

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
