# Peer review of "Mechanical Properties of PLA Specimens Obtained by Additive Manufacturing Process Reinforced with Flax Fibers"

_jcs, doi:10.3390/jcs7010027_

Round 1

Reviewer 1 Report

The manuscript presents the experimental analysis of the mechanical properties of 3D-printed flax fiber-reinforced PLA composites. The manuscript is carelessly written and needs to be clearly presented. The abstract is not clear and should be concise. The abstract should give a summary of the whole research. The materials and methods used in the study are not well described. The results are not discussed.

Following are the comments for the authors to improve the manuscript quality.

-          The contribution of this manuscript is not discussed or compared to what is reported in the literature. The introduction lacks studies that are related to this work. More studies are required in the introduction section to highlight the importance of this study further.

-          Poor language structure. The complete manuscript must be proofread and revised for English and grammar-related corrections. Examples of such mistakes are (not limited to):

o   Lines 19-21

o   Line 26

o   Line 28

o   Lines 148-149

o   Line 151 (In Figures 1)

o   Line 131 (In Table 1 is)

o   Lines 172-173: “The flax fibers are used to reinforce the 3D printed specimens. On the other hand, to manufacture the specimens, was chosen a biodegradable thermoplastic, PLA …”.

o   Line 182 “The process it began by …”

-          Justifications and references must be added to Lines 43-44.

-          Reference must be added to Lines 74-76.

-          Materials subsection (in section 2) must be added.

-          Materials and methods were not discussed adequately. The order of subsections should be revised, e.g., materials, printing, and mechanical tests.

-          The results MUST be compared with the pure PLA, e.g., without reinforcing, to show the benefits/drawbacks of adding  the fibers.

-          Line 128, “In contrast, the mechanical properties of NFPCs …”. Which is the contrast?

-          Table 1: the levels of factor A should be no treatment and NaOH treatment.

-          Table 1: Factor C units.

-          Tables layout

-          Lines 164-170 must be rewritten in a paragraph.

-          Lines 206-211 must be rewritten in a paragraph.

-          Lines 223-226 must be rewritten in a paragraph.

-          Lines 257-261 must be rewritten in a paragraph.

-          Line 175: “…. good adhesion between.”. Between what?

-          Table 1 defines the flax fiber as a percentage, while it was defined in Lines 183-184 as a number of strands.

-          How do the authors ensure that the nozzle will not hit the fibers and the molds? Figures should be added to show the reinforcing process.

-          What is the black-colored thing in Fig 3? It looks like carbon fibers sheet!

-          How do the authors ensure the used tapes will not influence the mechanical properties?

-          Line 228: “The tensile test took place in the laboratory of structures and strength of materials.” and Line 229: “First, the machine and computer were turned on and the data regarding the specimen …”. What is the benefit of reporting such sentences?

-          Is ANOVA an optimization tool? Lines:” The relative importance of the control factors for tensile and flexural strength needs to be identified more precisely using ANOVA analysis to determine the optimal combinations of control factor levels.”

-          Inconsistency in citing figures in the paragraph, e.g., Line 371 “figure 10”.

-          There are no discussions in the “4. Discussion” section. There are only statistical results. Sections 3 and 4 must be combined in one section, “Results”. A scientific discussion section must be added for the obtained results from the mechanical, material, thermal, etc. perspectives’.

-          Findings of this work must be discussed in reference to the previous studies.

-          The conclusion should be rewritten with bullets.

Author Response

The authors would like to acknowledge the careful revision done by the Reviewer, as well as his/her precious suggestions and comments. Also, we would like to thank the Editor for the opportunity that was given to us regarding the improvement of the original version of the manuscript (MS) and the further resubmission of the improved MS. Hence, we hope that this new version of the MS is worth publishing in the Journal of Composites Science. All the changes are highlighted in red color.

The manuscript presents the experimental analysis of the mechanical properties of 3D-printed flax fiber-reinforced PLA composites. The manuscript is carelessly written and needs to be clearly presented. The abstract is not clear and should be concise. The abstract should give a summary of the whole research. The materials and methods used in the study are not well described. The results are not discussed.

Thank you very much for your suggestion, we agree with it. So, we change the abstract following your recommendations. Please, see the new version of abstract which is highlighted in red color.

Following are the comments for the authors to improve the manuscript quality.

-          The contribution of this manuscript is not discussed or compared to what is reported in the literature. The introduction lacks studies that are related to this work. More studies are required in the introduction section to highlight the importance of this study further.

We are grateful for your advice, so, we improve the introduction by adding more detailed information based on new and updated references. Please, read the new introduction.

-          Poor language structure. The complete manuscript must be proofread and revised for English and grammar-related corrections. Examples of such mistakes are (not limited to):

o   Lines 19-21

o   Line 26

o   Line 28

o   Lines 148-149

o   Line 151 (In Figures 1)

o   Line 131 (In Table 1 is)

o   Lines 172-173: “The flax fibers are used to reinforce the 3D printed specimens. On the other hand, to manufacture the specimens, was chosen a biodegradable thermoplastic, PLA …”.

o   Line 182 “The process it began by …”

Thank you very much for your recommendations. We improve most of the English text enlisting the support of a professional in English translation.

-          Justifications and references must be added to Lines 43-44.

Thank you very much for the comment. Indeed, we must prove our statement with a reference. We have read this statement in the review paper (reference 4). We highlighted the statement in MS with red color.

-          Reference must be added to Lines 74-76.

Thank you very much for the comment. Indeed, we must prove our statement with a reference. We have read this statement in the paper which reference is 21. We highlighted the statement in MS with red color.

-          Materials subsection (in section 2) must be added.

Thanks for your suggestion. We add the sub-section “2.2 Materials”. It is highlighted in the MS with red color.

-          Materials and methods were not discussed adequately. The order of subsections should be revised, e.g., materials, printing, and mechanical tests.

We agree with your comment. For this reason, we have made major changes in the "Materials and Methods" section. Thus, we believe that most of the doubts or failures are now clarified in the MS.

-          The results MUST be compared with the pure PLA, e.g., without reinforcing, to show the benefits/drawbacks of adding  the fibers.

Thank you very much for your attention. In fact, we did the experimental tests for pure PLA, however, we forgot to include that information. In the revised MS version this information is included and commented on Lines 287-289 is referred to these experimental tests. In Figures 7 and 8 the curve of “Test 19” is the tests implemented for specimens without fibers.

-          Table 1: the levels of factor A should be no treatment and NaOH treatment.

Thank you for your suggestion, we did this change in Table 1.

-          Table 1: Factor C units.

Thanks, we add the units (ºC).

-          Tables layout

-          Lines 164-170 must be rewritten in a paragraph.

Thank you for your suggestion, we did this change,

-          Lines 206-211 must be rewritten in a paragraph.

Thank you for your suggestion, we did these changes.

-          Lines 223-226 must be rewritten in a paragraph.

Thank you for your suggestion, we did this change,

-          Lines 257-261 must be rewritten in a paragraph.

Thank you for your suggestion, we did this change,

-          Line 175: “…. good adhesion between.”. Between what?

Thank you for calling our attention, it was our mistake that we corrected.

-          Table 1 defines the flax fiber as a percentage, while it was defined in Lines 183-184 as a number of strands.

Thank you, it was a mistake that we corrected.

-          How do the authors ensure that the nozzle will not hit the fibers and the molds? Figures should be added to show the reinforcing process.

Thank you for your question. In fact, it is very difficult to guarantee that the nozzle never touches the fibers. We programmed that nozzle must have 1mm of the distance between layers, so, we think that the nozzle did not hit the fibers. Randomly, we observed closely (by eyes) the printing process and we never see the nozzle touching the fibers.

-          What is the black-colored thing in Fig 3? It looks like carbon fibers sheet!

Thank you for your question. The black color is due to the used PLA which had this color.

-          How do the authors ensure the used tapes will not influence the mechanical properties?

Thanks for the question. We used the tape for two main reasons, first, we used it to guarantee the special position (direction and distance between the strands) of the fibers and to impose a small pre-load to fibers and according to reference 43 (Rout, et. Al, 2001) the pre-load in the fibers could improve the mechanical properties.

-          Line 228: “The tensile test took place in the laboratory of structures and strength of materials.” and Line 229: “First, the machine and computer were turned on and the data regarding the specimen …”. What is the benefit of reporting such sentences?

Thank you for your question, we agree with you and rewrite the sentence, to be more objective and synthetic.

-          Is ANOVA an optimization tool? Lines:” The relative importance of the control factors for tensile and flexural strength needs to be identified more precisely using ANOVA analysis to determine the optimal combinations of control factor levels.”

Thanks for your comment. It was our mistake; we already correct it in the MS. We used the ANOVA analysis to determine de contribution of each control factor for a determined objective.

-          Inconsistency in citing figures in the paragraph, e.g., Line 371 “figure 10”.

Thank you for calling our attention, we already correct this mistake in MS.

-          There are no discussions in the “4. Discussion” section. There are only statistical results. Sections 3 and 4 must be combined in one section, “Results”. A scientific discussion section must be added for the obtained results from the mechanical, material, thermal, etc. perspectives.

Thanks for your suggestion, we merge section 3 and 4, creating one section that we call “3. Results and discussion”. We also improve the “discussion” and comments about the obtained results.

-          The conclusion should be rewritten with bullets.

Thank you for your suggestion. We rewritten the conclusion with bullets.

Reviewer 2 Report

There are some weaknesses through the manuscript which need improvement. Therefore, the submitted manuscript cannot be accepted for publication in this form, but it has a chance of acceptance after a major revision. My comments and suggestions are as follows:

1- Abstract gives information on the main feature of the performed study, but some details about the conducted tests must be added.

2- First part of abstract (first two sentences) is general and can be removed.

3- Authors must clarify necessity of the performed research. Objectives of the study must be clearly mentioned in introduction.

4- The literature study must be enriched. In this respect, authors must read and refer to the following papers: (a) https://doi.org/10.1038/s41598-022-05005-4 (b) https://doi.org/10.1016/j.mtcomm.2022.103402

5- Details of calculation must be added to the manuscript.

6- All figures must be illustrated with a high quality.

7- Some sentences need appropriate reference. Otherwise, they must be removed. For instance; “The polylactic acid (PLA) is one of the most used materials in additive manufacturing 16 processes”. And “Flax fiber is considered one of the strongest fibers”.

8- Why this particular material is used for this study.

9- The main reference of each formula must be cited. Moreover, each parameters in equations must be introduced. All formula must be presented by an identical font size. Please double check this issue.

10- Since fiber reinforced 3D-printed part is an interesting topic in current research programs, related issues such as fracture of 3d printed glass fiber composite, numerical modeling, optimization, and defect can be discussed with referring to relevant publications.

11- Standard deviation is the presented results must be discussed.

12- In its language layer, the manuscript should be considered for English language editing. There are sentences which have to be rewritten.

13- The conclusion must be more than just a summary of the manuscript. List of references must be updated based on the proposed papers. Please provide all changes by red color in the revised version.

Author Response

The authors would like to acknowledge the careful revision done by the Reviewer, as well as his/her precious suggestions and comments. Also, we would like to thank the Editor for the opportunity that was given to us regarding the improvement of the original version of the manuscript (MS) and the further resubmission of the improved MS. Hence, we hope that this new version of the MS is worth publishing in the Journal of Composites Science. All the changes are highlighted in red color.

There are some weaknesses in the manuscript which need improvement. Therefore, the submitted manuscript cannot be accepted for publication in this form, but it has a chance of acceptance after a major revision. My comments and suggestions are as follows:

1- Abstract gives information on the main feature of the performed study, but some details about the conducted tests must be added.

Thank you very much for your suggestion, we agree with it. So, we change the abstract following your recommendations. Please, see the new version of the abstract which is highlighted in red color.

2- First part of abstract (first two sentences) is general and can be removed.

The authors are grateful for the reviewer's suggestion. However, we consider that the first part of the abstract must have one or two sentences on background that should include the motivation(s) for the work. In this sense, we rewrite the first part of the abstract in a more concise way.

3- Authors must clarify necessity of the performed research. Objectives of the study must be clearly mentioned in introduction.

Thank you for the suggestion. Indeed, we agree with you, there is a lack of information about the objectives of the work. We change the introduction and now we clearly explain the main objectives of the work.

4- The literature study must be enriched. In this respect, authors must read and refer to the following papers: (a) https://doi.org/10.1038/s41598-022-05005-4 (b) https://doi.org/10.1016/j.mtcomm.2022.103402

The authors are very grateful for the suggestions of these references. We include them in the MS and they improve the quality of the report.

5- Details of calculation must be added to the manuscript.

Thanks for your comment. We added more information about the calculation that we used in our work.

6- All figures must be illustrated with a high quality.

Thank you for your observation. The figures that we used for the review process have a lower resolution than we intend to use in the final document because the document for review can be lither and easy to handle.

7- Some sentences need appropriate reference. Otherwise, they must be removed. For instance; “The polylactic acid (PLA) is one of the most used materials in additive manufacturing processes”. And “Flax fiber is considered one of the strongest fibers”.

Thank you for your comment. We agree with you add references that prove our sentences. As examples, we can refer to two references: https://doi.org/10.3390/polym13091407, https://doi.org/10.1016/j.compositesa.2005.10.011.

8- Why this particular material is used for this study.

Most of the authors are working in FabLab and we usually manufacture pieces in PLA, however, most of them cannot be used in practical applications due to the low strength of this. For this reason, we are interested to improve the mechanical strength of our pieces and, at the same time, use biodegradable fibers. The option for flax was due to two reasons, the first one is due to the fact that its mechanical properties are high, considering it is a natural fiber, and the second reason is related to the location of our work institution. This is in an agricultural region where there are some flax producers and the possibility of creating a composite with this material could be a new market for these producers in the future.

9- The main reference of each formula must be cited. Moreover, each parameter in equations must be introduced. All formula must be presented by an identical font size. Please double check this issue.

Thank you for your suggestion. We included the references we consulted for the formulas used in the work. We improve the formatting of equations.

10- Standard deviation is the presented results must be discussed.

Thanks for your comment. The discussion about the standard deviation was added in MS.

11- In its language layer, the manuscript should be considered for English language editing. There are sentences which have to be rewritten.

Thank you very much for your recommendations. We improve most of the English text enlisting the support of a professional in English translation.

12- The conclusion must be more than just a summary of the manuscript. List of references must be updated based on the proposed papers. Please provide all changes by red color in the revised version.

Thank you very much for your comments. We improved our conclusions. The references were updated based on the proposals papers. All changes were highlighted in red color.

Round 2

Reviewer 1 Report

The authors have significantly improved the manuscript.

Reviewer 2 Report

The paper has been improved and corresponding modifications have been conducted. In my opinion, the current version can be considered for publication.